



# Experimental investigation of the effect of wake steering on the noise emission of a commercial wind turbine

Thomas Duc[1,*] and Arthur Finez[1,*]

[1]ENGIE Green France, 6 rue Alexander Fleming, 69007 LYON, FRANCE
[*]These authors contributed equally to this work.

**Correspondence:** Thomas Duc (thomas.duc@engie.com) and Arthur Finez (arthur.finez@engie.com)

**Abstract.** Wake steering is a wind farm flow control (WFFC) strategy that involves intentionally misaligning the most upstream wind turbines to deviate their wakes away from the downstream wind turbines. This study investigates the acoustic implications of such a strategy. A novel acoustic setup was implemented, involving 24 ground-based sound level meters arranged on a circle around an industrial 2.2 MW wind turbine, positioned at tip-height distance. This configuration enables a fine angular resolution of 15°. Incoming wind conditions were monitored using both a nacelle-mounted plus a ground-based lidar. A test protocol closely aligned with IEC 61400-11-1 was followed to characterize the turbine's noise emissions under various yaw offset settings, ranging from -20° to +20° across a broad spectrum of flow conditions. A dedicated data cleaning and analysis procedure was developed to derive ground-level turbine noise directivity patterns. In the absence of yaw misalignment, the directivity patterns exhibit the typical two-lobe structure. However, slight but statistically significant asymmetries were also observed: the downstroke side was in average 0.6 dB(A) louder than the upstroke side, and the downwind side was 0.4 dB(A) louder than the upwind side. These specificities are not captured by most of the analytical models used by the profession. When yaw control was applied, a modest increase of approximately 0.6 dB(A) was observed in the estimated overall sound power level. The results from this innovative experiment confirm the fact that operators must consider other metrics than just power production when implementing WFFC on their projects, and that more advanced noise models are required for the development of a framework allowing multi-objective WFFC.

## 1 Introduction and context

Wind Farm Flow Control (WFFC) is a concept in which turbines within a wind farm are controlled in a coordinate way to accomplish a common objective, such as increase in farm power production, reduction in turbine loading, or active power control (Boersma et al., 2017). Two strategies are generally considered to achieve these goals : the axial induction strategy in which the upstream turbines are down-regulated to leave more kinetic energy for the downstream turbines and the wake steering strategy that consists in intentionally misaligning the upstream turbines to deflect the wake away from the downstream turbines. For both of them, the performance of the most upstream turbines is reduced so that the one of the downstream turbines is improved: the WFFC is successful when the downstream gain is compensating for the upstream loss and the overall power production is increased. WFFC has been an intense subject of research over the past decade with many full scale experiments



organized all over the world on commercial wind farms. The vast majority of these experiments were focused on the objective of improving the total energy production of a wind farm, while only a few of them tackled the issue of load reduction (Damiani et al., 2018; Dana et al., 2022) or active power control (Göçmen, 2016), .

Among the two strategies, wake steering is the one that has shown the higher potential for power improvements, with a track record of many full scale field tests showing significant energy gains (see e.g. Fleming et al. (2017, 2019, 2020, 2021);
Simley et al. (2021); Doekemeijer et al. (2021); Howland et al. (2022)). On the other hand, meaningful power gains seems more difficult to achieve through axial induction control, although a few full scale experiments were able to highlight performance improvements for specific wind sectors (van der Hoek et al., 2019; Bossanyi and Ruisi, 2021). When averaging over the full wind rose to get the total annual energy gains, those gains are drastically reduced and consequently this strategy might be more suitable for achieving load reduction and active power control (Boersma et al., 2017).

Another less widespread objective that can be achieved through WFFC is acoustic control of a wind farm (Nyborg et al., 2023). Wind turbines are controlled together to comply with the local legislation and make sure the total noise emissions of the farm remains below a given threshold, while trying to maximize the total energy production. This is usually accomplished using an axial induction strategy during which specific noise reduced operation (NRO) modes are activated according to a list of criteria (for instance time of day, period of year, incoming wind speed and direction sector). A practical example of an
implemented acoustic curtailment plan is presented in Figure 1. Note that it is usually defined using numerical simulations and simple noise source models assuming omnidirectional noise sources. A variation of $\pm 0.5$ dB(A) in the source power definition can be qualified as "small" from the acceptance point of view, since it is lower than 1 dB, the level variation threshold of human hearing. However, in critical situations, such variation can lead to significant changes in the curtailment plan associated with variations in Annual Energy Production (AEP) of the order of 1%, since the chosen curtailment plan must strictly comply with
local noise regulations.

An expert elicitation about WFFC realized a few years ago only ranked the noise reduction objective in second to last position (van Wingerden et al., 2020). However, in some countries where regulations and constraints are very restrictive, the topic of farm noise emissions has become increasingly important. As an example, the large majority of newly developed wind projects in France are now concerned with an acoustic curtailment, leading to an average 2.8% reduction in AEP (Willis, 2023).

The TWAIN research project aims at providing a framework for multi-objective WFFC: increase in power production, reduction in turbine loading, complying with grid requirements or environmental regulations (acoustics, wildlife and social consciousness). Therefore, understanding the impact of WFFC actions not only on turbine power and loads but also on other aspects such as farm noise emissions is critical to design the optimal strategies to be applied by the wind farm owner or operator.
In that scope, acoustic models must be improved and validated to bridge the gap with the current state of the art.

Noise impact studies dedicated to the permitting of industrial wind farm projects most often involve omnidirectional point noise sources located at the nacelle center of the wind turbines. However, several scientific studies challenge this omnidirectionality assumption. Acoustic measurements on full scale wind turbines consistently show a "noise dip" of a few decibels in the rotor plane (Oerlemans and Schepers, 2009; Buck et al., 2016; Okada et al., 2016), which is generally reproduced by more





| Time of day | Wind sector | Season | Turbine | Wind speed | | | | | | | | | | | | |
|---|---|---|---|---|---|---|---|---|---|---|---|---|---|---|---|---|
| | | | | 3 m/s | 4 m/s | 5 m/s | 6 m/s | 7 m/s | 8 m/s | 9 m/s | 10 m/s | 11 m/s | 12 m/s | 13 m/s | 14 m/s | 15 m/s |
| Night | North-East | Spring & Summer | WT1 | Mode 0 | Mode 0 | Mode 0 | Mode 2 | Mode 2 | Mode 2 | Mode 2 | Mode 2 | Mode 2 | Mode 2 | Mode 2 | Mode 2 | Mode 0 |
| | | | WT2 | Mode 0 | Mode 0 | Mode 0 | Mode 0 | Mode 1 | Mode 1 | Mode 0 | Mode 0 | Mode 0 | Mode 0 | Mode 0 | Mode 0 | Mode 0 |
| | | | WT3 | Mode 0 | Mode 0 | Mode 0 | Mode 2 | STOP | STOP | Mode 1 | Mode 1 | Mode 0 | Mode 0 | Mode 0 | Mode 0 | Mode 0 |
| | | | WT4 | Mode 0 | Mode 0 | Mode 0 | Mode 2 | Mode 2 | Mode 2 | Mode 0 | Mode 0 | Mode 0 | Mode 0 | Mode 0 | Mode 0 | Mode 0 |
| | | Fall & Winter | WT1 | Mode 0 | Mode 0 | Mode 0 | Mode 0 | Mode 4 | STOP | STOP | Mode 4 | Mode 0 | Mode 0 | Mode 0 | Mode 0 | Mode 0 |
| | | | WT2 | Mode 0 | Mode 0 | Mode 0 | Mode 0 | Mode 4 | STOP | STOP | STOP | Mode 0 | Mode 0 | Mode 0 | Mode 0 | Mode 0 |
| | | | WT3 | Mode 0 | Mode 0 | Mode 0 | Mode 2 | Mode 2 | Mode 4 | Mode 4 | Mode 4 | Mode 0 | Mode 0 | Mode 0 | Mode 0 | Mode 0 |
| | | | WT4 | Mode 0 | Mode 0 | Mode 0 | Mode 2 | STOP | STOP | Mode 4 | Mode 4 | Mode 0 | Mode 0 | Mode 0 | Mode 0 | Mode 0 |
| | South-West | Spring & Summer | WT1 | Mode 0 | Mode 0 | Mode 0 | Mode 0 | Mode 0 | Mode 0 | Mode 0 | Mode 0 | Mode 0 | Mode 0 | Mode 0 | Mode 0 | Mode 0 |
| | | | WT2 | STOP | STOP | STOP | STOP | STOP | STOP | Mode 2 | Mode 2 | Mode 2 | Mode 2 | Mode 2 | Mode 2 | Mode 2 |
| | | | WT3 | Mode 0 | Mode 0 | Mode 0 | Mode 0 | Mode 0 | Mode 0 | Mode 1 | Mode 1 | Mode 1 | Mode 0 | Mode 0 | Mode 0 | Mode 0 |
| | | | WT4 | Mode 0 | Mode 2 | Mode 2 | Mode 2 | Mode 2 | Mode 2 | Mode 2 | Mode 2 | Mode 2 | Mode 2 | Mode 2 | Mode 2 | Mode 2 |
| | | Fall & Winter | WT1 | Mode 0 | Mode 0 | Mode 0 | Mode 2 | Mode 2 | Mode 4 | Mode 4 | Mode 2 | Mode 2 | Mode 2 | Mode 2 | Mode 0 | Mode 0 |
| | | | WT2 | Mode 0 | Mode 0 | Mode 0 | STOP | STOP | STOP | Mode 4 | Mode 4 | Mode 2 | Mode 2 | Mode 2 | Mode 0 | Mode 0 |
| | | | WT3 | Mode 0 | Mode 0 | Mode 0 | Mode 0 | Mode 0 | Mode 0 | Mode 0 | Mode 0 | Mode 0 | Mode 0 | Mode 0 | Mode 0 | Mode 0 |
| | | | WT4 | Mode 0 | Mode 0 | Mode 0 | Mode 2 | Mode 2 | Mode 2 | Mode 2 | Mode 2 | Mode 2 | Mode 1 | Mode 1 | Mode 0 | Mode 0 |

**Figure 1.** Typical acoustic curtailment applied in a wind farm. The wind farm control is applied as a look-up table giving the NRO mode to be followed by each turbine as a function of several environmental criteria. The turbines are supposed to be operated in full power mode outside the conditions provided in this lookup table (e.g. during daytime).

advanced analytical models. On the contrary, complex multi-lobe structures as well as upwind/downwind and upstroke/down-stroke asymmetries are observed but not fully captured by those models.

Similarly, while axial induction control has been used for years to implement acoustic curtailment plan, the effect of wake steering on noise emissions remains an open question. Only two full scale experiments related to this topic were found in the literature, with one pointing toward a reduction of the noise emission due to the application of yaw misalignment (Hamilton et al., 2021) and the other showing no hearable impact (Bonsma et al., 2019). Since wake steering has already demonstrated positive energy gains and begins to be deployed on many wind farms (Harrison et al., 2025), there is a need to quantify more accurately its consequences on acoustics.

This paper introduces a novel field campaign designed specifically to study the effect of turbine yaw control on sound magnitude and directivity. A commercial wind turbine has been fully surrounded by 24 microphones in order to measure with a high resolution the directivity pattern of its noise emissions and track how it is changed when yaw misalignment is applied. In the following the term "directivity" refers to the ground trace of the horizontal noise directivity, as measured by a circle of microphones around the turbine.

The field campaign, described in Sect. 2, was prepared following the recommendations of the IEC 61400-11 standard as close as possible (IEC et al., 2012), and therefore terms and notations adopted in this article are mostly inspired from this reference. However, due to the extensivity of the setup, some adjustments to the standard procedure had to be applied. Those are detailed in Sect. 3 along with the methodology applied to process the large dataset recorded during the campaign. Section





4 provides the analysis of noise directivity pattern and effect of yaw misalignment. The goal of this experiment is to provide
a high quality dataset that can be used within the TWAIN project to improve acoustics models; Sect. 5 will thus conclude this
paper and offer perspectives about future developments realized in the scope of the project.

## 2    Description of the field campaign

The field campaign was realized from 25 to 28 March 2024 on a commercial wind turbine (diameter D = 110 m, nominal power
P = 2.2 MW and hub height HH = 80 m) in a farm operated by ENGIE Green and located in the northern region of France.
This section gives a summary of the experimental setup, the procedure followed and the wind conditions recorded during the
test.

### 2.1    Experimental setup

Figure 2 recaps the experimental setup that was implemented for the campaign. The 24 sound level meters (SLM) surrounding
the turbine were located every 15° at a distance of approximately one turbine tip height (135 m) from the turbine. Distance
to the tower base was measured thanks to laser pointer and cross checked using their GPS position: the values obtained for
laser measurement ranged from 132.5 m to 139.4 m which fulfills the ± 20%, ± 30 m tolerance criteria of the IEC standard
(IEC et al., 2012). Angular difference between two successive SLM was evaluated using their GPS coordinates and varied
from 11.6° to 18.2°. The distance from the ground-based Windcube lidar to the turbine was also estimated based on its GPS
coordinates and found to be roughly 330 m.

The site is very simple and flat, and the calculated terrain inclination angle $\phi$ for all SLM position varied between 29°
and 31.5°, well within the 25° - 40° range preconized by the IEC standard. The surrounding terrain mostly consists of wheat
crops. In late March, at the time of the experiment, those crops were reaching a height of about 30 cm. In order to prevent any
disturbance from the noise caused by the wind on those crops, they were cut for a diameter of roughly 3 m around each SLM
position. The Fig. 3 shows one SLM together with controlled turbine in the background.

SCADA data for the misaligned turbine was collected during the test at 1 second resolution for the most important variables:
active power, wind speed, vane angle, nacelle position, rotor and generator speed, pitch angle. Almost all acoustics sensors
were removed each night and reinstalled every morning at the same location. They were checked for time synchronization and
calibration before new measurements were launched. Both the overall sound pressure levels and 1/3-octave bands spectra were
recorded with a sampling period of 1 second. Measurements from lidars were also done with the same temporal resolution.

### 2.2    Test procedure

In order to study the effect of yaw misalignment on wind turbine noise in similar atmospheric conditions, a specific measure-
ment routine was designed for the campaign. The yaw set point parameter $\beta$ is defined as the angle between the nacelle axis
and the incoming main flow direction.The turbine was sequentially yawed from $\beta$ = -20° to $\beta$ = +20° by step of 10°, staying in
each position for 40 minutes. Between two successive positions, background noise and turbine noise measurements were also



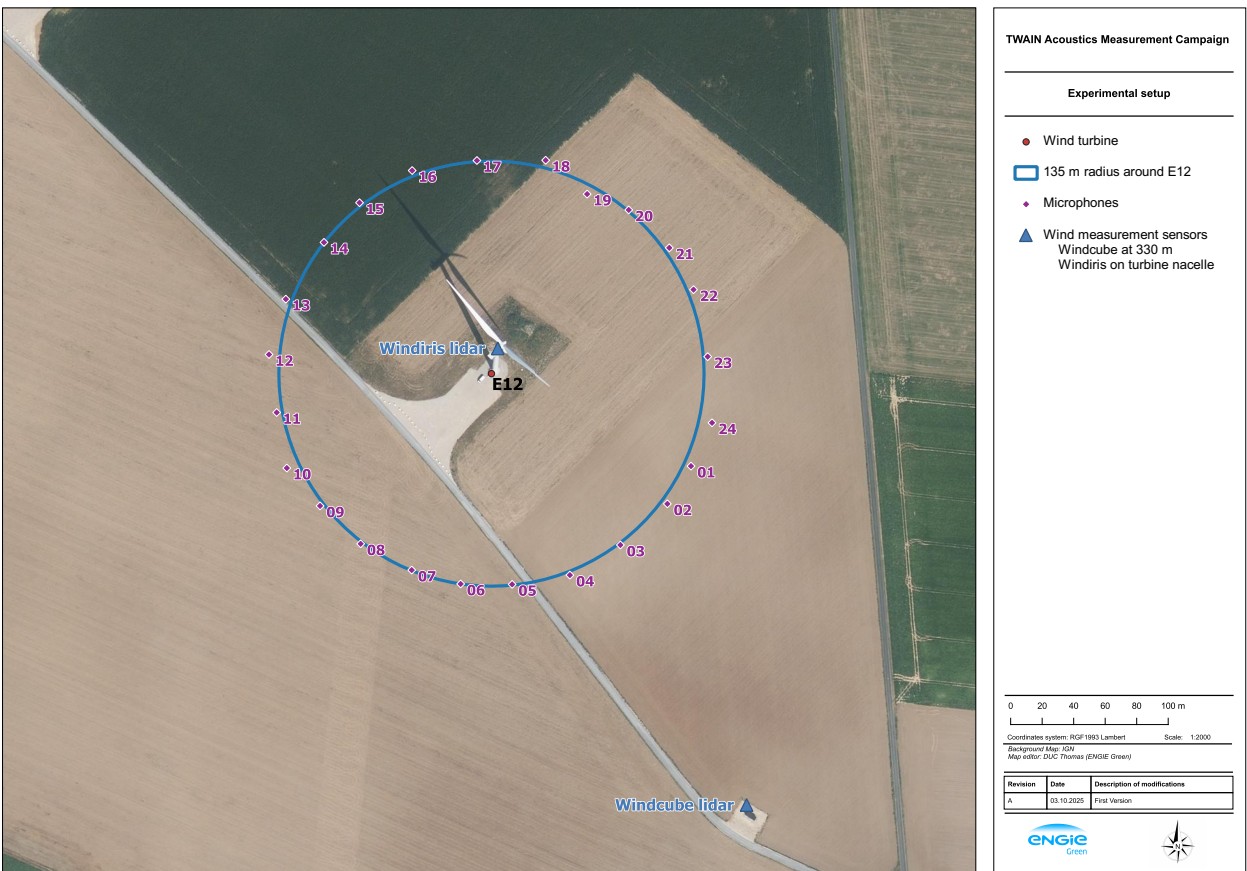

**Figure 2.** Experimental setup of the TWAIN acoustic measurement campaign. External wind sensors include a the ground-based Windcube lidar located at 330 m from the turbine and a nacelle-mounted Windiris lidar. The 24 microphones were placed approximately 1 total height of the turbine (135 m) from the tower base

conducted with $\beta = 0°$, each of them lasting 10 minutes. Finally, a period of 3 minutes was considered for each transition to give some time for the turbine to move or restart and let the flow reach its equilibrium for the new position. The full routine, planned to last almost 4.5 hours can be found in Tab. 1.

In practice, this routine could not be programmed in the control software and the turbine was manually switched between each state by an operator. The periods in each position were thus approximately followed but were maintained at least as long as indicated above. Overall, the fulfillment of one full round of measurement lasted a bit longer than originally planned. During the data post-processing, the SCADA 1 second timeseries was visually inspected to identify precisely the moments of switching from one phase to another. In operation phases, the target turbine was settled to the full power mode, i.e. without any NRO mode implemented.





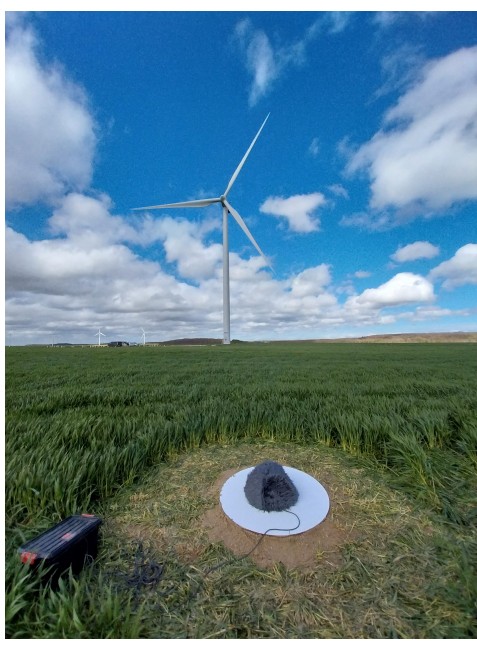

**Figure 3.** Picture of one SLM with E12 wind turbine in the background. The wheat crops around the SLM were cut to prevent noise disturbance. Following the recommendations of the IEC standard (IEC et al., 2012), all microphones are installed on a woodboard of 1 m diameter and covered by two windscreens.

The two nearest turbines in the farm (WT10 and WT11 located at respectively 345 and 770 m at the North-West of WT12) were shut down at the beginning of each round of measurement and maintained stopped for the full duration of the test to avoid

disturbing the test and record data with a satisfactory signal to noise ratio. The closest neighboring turbine still in operation was located more than 1 km away. A person was also constantly present on site to keep a track record of the noisy events (such as a plane or car passing by) so that they could be properly considered and filtered afterwards.

## 2.3    Wind conditions recorded during the test

The experiment consisted of 6 rounds of measurements that were conducted over the 4 days : one on Monday 25[th] and on

Tuesday 26[th], and two on Wednesday 27[th] and Thursday 28[th]. Round 2, 4 and 6 could not be realized completely due to the appearance of rain in the afternoon of the 26[th] and 27[th] and of strong and highly turbulent winds risking to endanger the turbine during yaw misalignment conditions in the late afternoon of the 28[th]. In total, a little more than 21 hours of measurements were recorded over the four days (4.58 h on day 1, 2.35 h on day 2, 6.13 h on day 3 and 8.07 h on day 4).

The wind conditions observed during the campaign are represented on Fig. 4. Very different conditions were experienced

each day : from low to medium wind speeds during the first two days to higher wind speeds on the two last ones. The IEC standard requires a minimum number of 10 samples in each bin. As can be seen on Fig. 5, representing the raw capture matrix derived from the SCADA data recorded during the 6 rounds of the test, this criteria is fulfilled for a large number of wind





**Table 1.** Description of the measurement routine followed during the test. Each phase is planned to last 40 minutes, the background noise and turbine noise measurement at $\beta = 0°$ are spread in 4 steps of 10 minutes each to be recorded closer to the misaligned conditions.

| Phase | Yaw offset $\beta$ (°) | WT operation | Duration |
|---|---|---|---|
| Turbine noise measurement | -20 | Operating | 00:40:00 |
| Background noise measurement | -20 | Stopped | 00:10:00 |
| Transition | Toggling -20 -> 0 | Starting | 00:03:00 |
| Turbine noise measurement | 0 | Operating | 00:10:00 |
| Transition | Toggling 0 -> +20 | Operating | 00:03:00 |
| Turbine noise measurement | 20 | Operating | 00:40:00 |
| Background noise measurement | 20 | Stopped | 00:10:00 |
| Transition | Toggling +20 -> 0 | Starting | 00:03:00 |
| Turbine noise measurement | 0 | Operating | 00:10:00 |
| Transition | Toggling 0 -> -10 | Operating | 00:03:00 |
| Turbine noise measurement | -10 | Operating | 00:40:00 |
| Background noise measurement | -10 | Stopped | 00:10:00 |
| Transition | Toggling -10 -> 0 | Starting | 00:03:00 |
| Turbine noise measurement | 0 | Operating | 00:10:00 |
| Transition | Toggling 0 -> +10 | Operating | 00:03:00 |
| Turbine noise measurement | 10 | Operating | 00:40:00 |
| Background noise measurement | 10 | Stopped | 00:10:00 |
| Transition | Toggling +10 -> 0 | Starting | 00:03:00 |
| Turbine noise measurement | 0 | Operating | 00:10:00 |
| Transition | Toggling 0 -> -20 | Operating | 00:03:00 |

speeds bins and all yaw positions thanks to those very favorable wind conditions. Consequently, the minimum wind speed range recommended by the standard, 8 to 13.5 m/s (corresponding to 0.8 to 1.3 times the wind speed at 85 % of the turbine

rated power), was extended to 6 - 13.5 m/s when analyzing the results.

The wind direction varied from East the first day to South-West the last day, meaning that the turbine was never affected by any wakes during the measurement. Turbulence and shear conditions were also very variable depending on the day, with a high shear and a low turbulence observed at the beginning of the week and on the contrary a highly turbulent and low sheared flow noticed at the end of the test.

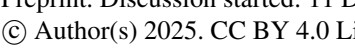



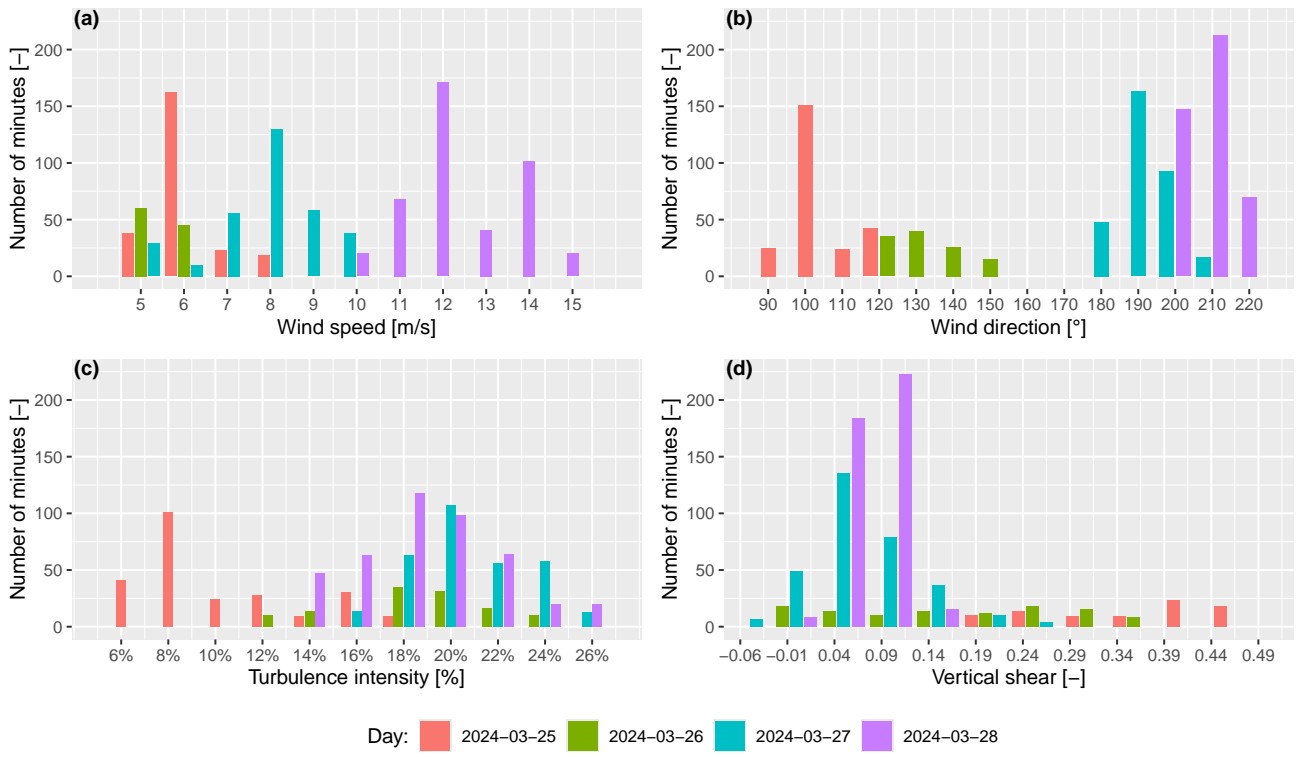

**Figure 4.** Wind conditions recorded during the field campaign. Histogram of occurrences of wind speed (a), wind direction (b), turbulence intensity (c) and wind shear (d) for each day of measurement.

## 2.4 Data availability

One of the biggest challenge of a field campaign implicating as many sensors as this one is to maintain the availability of all of them during the periods of measurement. As can be seen on Fig. 6 and Fig. 7, representing the availability of wind related and acoustic sensors, respectively, this challenge was practically met.

Most of the SLMs were also fully available during the test. Unfortunately some sensor defaults introduced a loss on data in some conditions. SLM #04 shows only a 10 % availability because it was only possible to record data every 10 seconds instead of 1 second due to a logger issue. SLM #07 reports a reduced availability since it was not installed during the second day of measurement. The IEC standard also requires an analysis of 1/3-octave bands between 20 Hz and 10 kHz. A full dataset was obtained for 18 SLMs but 1/3-octave information was missing for SLMs #09 and #18, and incomplete for #04, #10, #16, #21 and #22.

Despite these few inconveniences, the quantity of data collected during the 4 days and 6 rounds of measurement can be considered as satisfactory and ready to be processed for the derivation of the apparent sound power level, as is described in the next section.



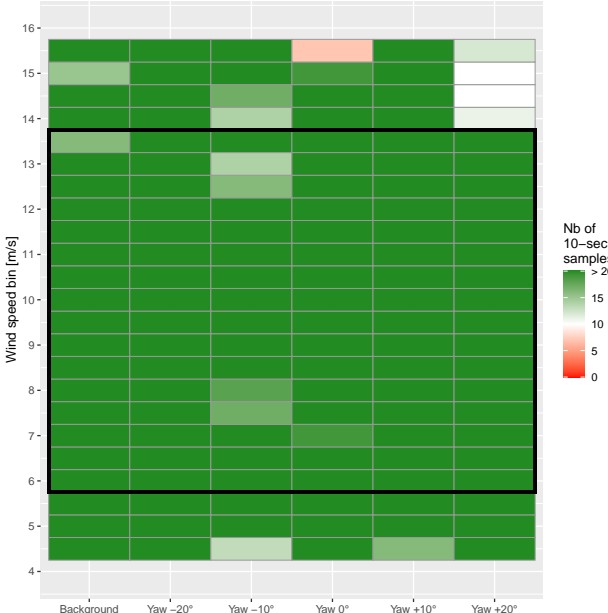

**Figure 5.** Raw capture matrix for the TWAIN acoustic campaign. The number of 10-seconds sample is shown for each wind speed bin and each yaw misalignment case. The black rectangle represents the wind speed range of interest between 6 and 13.5 m/s.

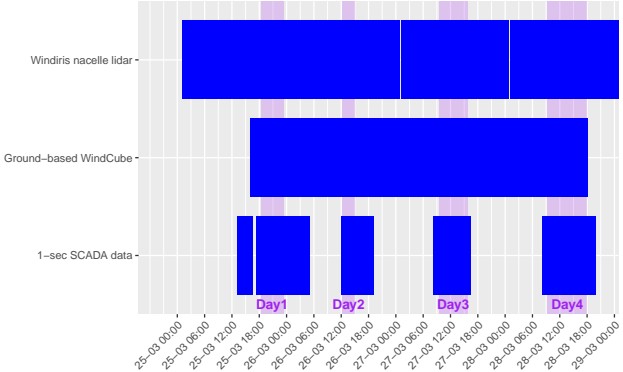

**Figure 6.** Availability of related wind sensors during the TWAIN acoustic campaign. The shaded purple vertical bands indicate active measurement for each day.

## 3 Data processing

The IEC 61400-11 standard defines a procedure for the processing of data recorded during an acoustic noise measurement
campaign: aggregation of data into 10-seconds chunks, time synchronization between acoustics and wind related sensors, estimation of ambient wind speed, normalization of 1/3-octave band spectra and correction for secondary wind screen, sorting





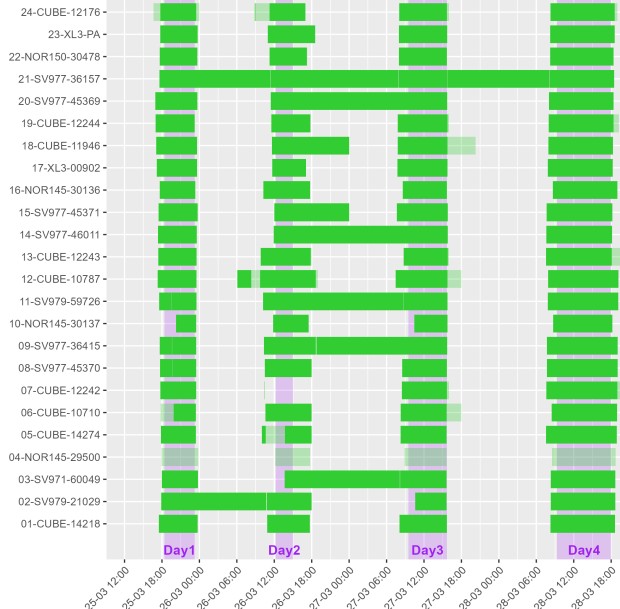

**Figure 7.** Availability of overall sound pressure level data for all SLMs during the TWAIN acoustic campaign. A transparent band (e.g. for SLM #04) corresponds to a reduced availability. The shaded purple vertical bands indicate active measurement for each day.

into bins, computation of average and standard deviation acoustic values on bin centres and computation of apparent sound power spectra through the difference between the total noise and background noise values.

However, this procedure corresponds to the case of a single SLM placed directly downstream of a turbine aligned with the wind direction. Due to the novelty of this campaign, implicating a large number of SLMs and the application of a yaw misalignment to the turbine, some adjustments to the standard procedure were necessary.

## 3.1 Acoustic data cleaning

The process described above must be performed on a cleaned dataset. Despite the effort made every day during the campaign to calibrate and check the time synchronization of each SLM before launching a new round of measurement, a few issues were noticed when analyzing the raw acoustics data. Considering the large quantity of data to handle (more than 507 hours when accounting for the 24 microphones), specific post-processing techniques were designed to automatically filter and correct the dataset.

### 3.1.1 Time synchronization

The biggest issue observed when looking at the data was related to the time synchronization of the acoustic signals. Indeed, during the campaign each SLM was recording data using its own internal clock. Although each clock was verified at the beginning of a new measurement, it proved not enough to ensure a proper synchronization of the signals and time shifts were

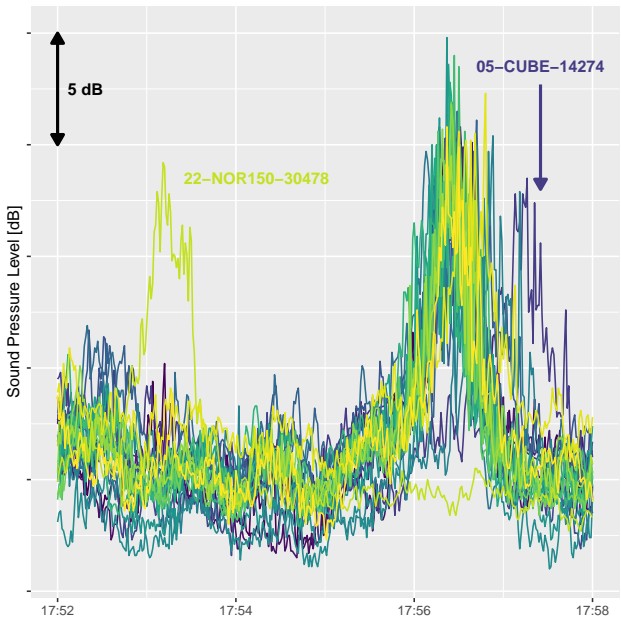

**Figure 8.** Illustration of a time sync issue during day 4. The passing of plane increases the sound pressure level of all SLMs between 17:56 and 17:57. This is captured by all of them at more or less the same time, except for SLM #22 which seems in advance and SLM #05 which is delayed.

found when inspecting the recorded time series. These time shifts were seen to be dependent on both the day and the sensor. Figure 8 shows an example in which two microphones clearly seem out of sync with respect to the rest of the fleet.

In order to massively identify and correct those time shifts, the following methodology was applied for each day of mea-
surement.

1. For each pair of SLMs, a cross-correlation curve is obtained by sliding the time signals by step of 1 second over ± 600 seconds.

2. The time delay that corresponds to the maximum of this cross-correlation curve is inserted into a matrix of time shifts that stores the most likely delay for each couple of SLMs for a given day of measurement.

3. A unique time shift per SLM is deduced by solving a linear optimization problem.

4. Finally, the acoustic signals are resynchronized with the turbine SCADA clock by using another cross correlation analysis with the electrical power signal (and taking advantage of the background noise measurement periods when the power suddenly drops to 0 kW with a an immediate reduction of its noise emissions).

The last step of the methodology is allowed by the fact that the subject of the study is focused on energetical aspects rather
than on acoustic propagation time. The distance between the main acoustic sources (the blade outer part for aerodynamic noise





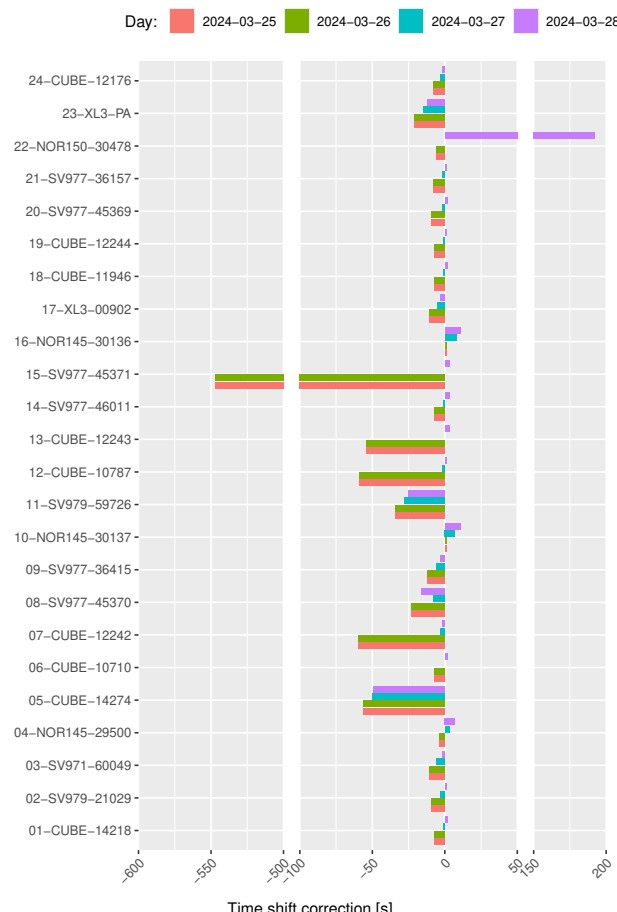

**Figure 9.** Time shift corrections applied for each microphone and each day of measurement. 0 seconds represents a perfect synchronization with the SCADA, a positive (resp. negative) value is found when the microphone data is ahead (resp. delayed) with respect to the SCADA.

and the nacelle for mechanical noise) is at most around 300 m, so the dispersion of acoustic propagation times remains on the order of a second. Given that the data is then aggregated into 10-seconds chunk for the rest of the IEC analysis, the uncertainty related to this correction can be considered negligible.

A summary of calculated time shift for all SLMs and all day of measurement is proposed on Fig. 9. It can be seen that the
vast majority of the introduced corrections are within $\pm$ 15 seconds. Nonetheless, significant delays were applied for some SLMs: -547 seconds were found for SLM #15 on days 1 and 2, and +192 seconds were obtained for SLM #22 on day 4 in accordance with the visual observations of Fig. 8.

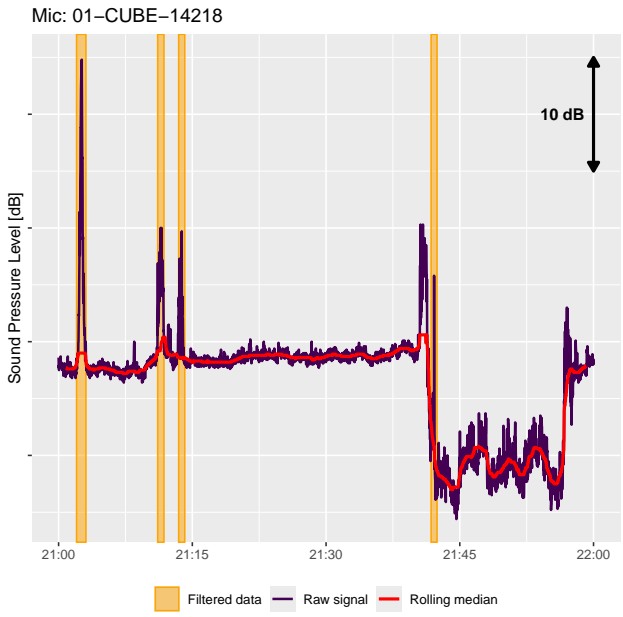

**Figure 10.** An example of automatic outlier detection for SLM #01. The rolling median is calculated from the raw signal and periods of $\pm$ 20 seconds above the + 10 dB threshold are detected and removed.

### 3.1.2 Noisy events and outliers removal

The requirements of the IEC standard indicate that periods with intermittent background noise shall be omitted from the analysis. As already explained in Sect. 2 above, an operator was always present onsite during the period of measurement to take note of the time when noisy events were heard (aircrafts, cars, agricultural machineries, ...). The exact starting and ending times of the events were readjusted after a visual inspection of the acoustic signals and the corresponding periods were filtered accordingly.

However, it was observed during the visual examination that a few abnormal data could not be related to any recorded noisy events. Those random outliers are characterized by sudden peaks in the acoustic signals. To automatically detect and remove those points, a $\pm$ 50 seconds running median of the acoustic signal is calculated. Every time the original signal goes 10 dB above the running median, an outlier is marked and data within $\pm$ 20 seconds around this point is filtered. The full process is illustrated on Fig. 10. A total of 1963 outliers were identified thanks to this method for the 24 SLMs over the 4 days of the campaign, with disparate behaviors depending on the sensors.

### 3.1.3 Final adjustments and aggregation

In addition to these global cleanings applied to all microphones, a few individual adjustments were also implemented on some SLMs to correct specific issues that were detected.





– SLM #06 showed a very low noise level for a few hours during day 1 despite a normal calibration procedure. This behavior was not repeated the following days of the campaign and could not be explained by the SLM manufacturer other than by a bad connection. Since variations of its signal were consistent with the ones observed at its neighbors, it was finally decided to correct the measures on day 1 by applying a constant +29.5 dB offset, calculated to fit the mean difference observed with the two closest microphones during the 3 other days of the campaign.

– SLM #21 only recorded 1/1-octave band instead of 1/3-octave data. The 1/3-octave band was obtained through a linear interpolation of the 1/1-octave band data, with the additional constraint of making sure that the sum of energy of the 3 calculated consecutive 1/3-octave bands was consistent with the measured level for the corresponding 1/1-octave band.

After those final corrections, the data preprocessing steps defined in the IEC procedure were followed. All signals were first aggregated into 10-seconds chunks, then the 1/3-octave data were normalized to the measured overall sound pressure level and the correction for the secondary wind screen applied.

### 3.2 Identification of reference wind conditions

The data reduction procedure described in the IEC standard stipulates that the data must be binned by wind speed. The reference wind speed to be used for the binning is deduced from the active power signal by inverting the turbine power curve. The (adjusted) nacelle anemometer and external wind sensors are used as alternative source of data when the power curve cannot be used (i.e. nominal power reached or turbine stopped for background measurement).

However this methodology is only valid for a turbine that is aligned with the wind. Indeed, the performance of a steering turbine is reduced and no longer follows the expected power curve. Measurements from nacelle anemometers are also known to be unreliable when yaw misalignment is applied on a turbine (Kanev, 2020; Astolfi et al., 2023). It was thus necessary to adapt the IEC methodology to account for the specificity of this test.

For this field campaign, the reference wind speed signal was calculated as follows.

1. When the turbine was operating with no yaw misalignment, and the wind speed was within the allowed range for interpolation prescribed by the standard, the active power signal and the power curve were used. The power curve considered for the interpolation was calculated thanks to the nacelle lidar data over 4 months of measurements from 7 February to 2 June 2024 and according to the procedure of the IEC standard 61400-12-1 (IEC et al., 2017).

2. When the turbine was operating with no yaw misalignment but the wind speed was outside allowed interpolation range, the nacelle anemometer, adjusted with a factor $\kappa_{nac} = 0.984$, was chosen.

3. During background noise measurement or when yaw misalignment was applied to the turbine, the measurement of the nacelle lidar at 300 m upstream (2.7 rotor diameters), adjusted with a factor $\kappa_z = 0.931$, was selected.

This sensor was favored over the ground based lidar because its wind speed measurements were better correlating with the wind speed derived from the active power signal in step 1 above.





The high frequency measurement of such a lidar is only realized along line of sights (LOS) of the sensor. Data from the 4 LOS must be averaged over a certain time period and combined together to derive the hub height wind speed estimation (Mazoyer and Boquet, 2016). Unfortunately, it proved that the 10-seconds resolution was too short a period to properly achieve this reconstruction: the resulting wind speed signal was too noisy and did not correlate well with the SCADA data. While complex algorithms exist to assess the incoming wind field from high frequency lidar measurements (Raach et al., 2014; Borraccino et al., 2017; Guillemin et al., 2018), another solution was set up instead, for the sake of simplicity. For each day,

a constant advection time was calculated using the mean wind speed observed during the period of record and the distance between the upstream lidar measurement range and the turbine. This advection time was used to shift temporally the lidar data and synchronize them with the rotor plane. Finally, wind speed measurements were averaged for every 10-seconds chunks considering a rolling window of 1 minute.

While wind direction is only considered in the IEC standard as a filtering variable, it is for this campaign a quantity of high

interest. The experimental setup being fully symmetric, the relative position of each microphone with respect to the turbine is determined based on the absolute wind direction. Likewise the nacelle anemometer, the turbine wind vane cannot be trusted under yaw misalignment (Rott et al., 2023). To ensure a consistent measure of wind direction during the full test, one of the external sensors had to be used and their correlations to the turbine data at 0° yaw were analyzed. The ground based lidar proved to have the most consistent measure of wind direction and its signal at turbine hub height was therefore selected.

### 3.3 Derivation of apparent sound power level

Once noise data have been cleaned, and reference wind conditions identified, both datasets can be combined and sorted into bins for both background and total noise to deduce the apparent sound power of the turbine. The following subsections describes how the IEC standard procedure was adapted to account for the particular setup of this field campaign. For the sake of clarity, Tab. 2 recaps the variables and symbols used in this section.

#### 3.3.1 Identification of background noise

With 24 microphones comes as many estimation of background noise. Figure 11 displays the evolution of background noise measured by the 24 sensors as a function of wind speed. All of them show a clear increasing trend with more or less the same slope. However at a given wind speed significant differences can be noticed, reaching up to approximately 4 dB when considering the two extreme microphones. These gaps can be explained by the variety of local environment around each sensor.

For example, SLM #04 installed closer to the substation than SLM #19 which is located in the middle of the field might not show the same background noise level (see Fig. 2). Difference in crops cutting around each microphone might also have some influence, especially at high wind speeds.

Due to the large disparity of background noise observed at the site, it was not possible to extract a single background noise value valid for all SLMs as a function of wind speed. Instead the approach proposed by Hamilton et al. (2021) was followed,

with a specific background noise dataset binned by wind speed calculated for each SLM.





**Table 2.** List of symbols considered in the data reduction procedure. The first section defines the physical variables used as subscripts, the second the data partition used in the procedure, and the third the acoustic-related quantities calculated. The subscript letter $c$ is taken from the IEC 61400-11 standard and refers to the background noise correction.

| Symbol | Variable |
|---|---|
| $i$ | 1/3-octave band |
| $v$ | Wind speed bin |
| $m$ | Sound Level Meter (SLM) |
| $r$ | Relative position |
| $\beta$ | Yaw offset category |
| $k = \{v; \beta; r; m\}$ | Partition defining a given wind speed, yaw category, relative position and SLM |
| $N_k$ | Number of data points in partition $k$ |
| $l = \{v; \beta; r\}$ | Partition defining a given wind speed, yaw category, relative position |
| $N_l$ | Number of data points in partition $l$ |
| $M_l$ | Number of SLM $m$ that can be combined together in partition $l$ |
| $w_k$ | Weight of each microphone within partition $l$ |
| $L_{Vc_{i,k}}, u_{c_{i,k}}$ | 1/3-octave band background-corrected sound pressure level (SPL) and corresponding uncertainty |
| $L_{Vc_k}, u_{c_k}$ | Background-corrected SPL and corresponding uncertainty (aggregation over all 1/3-octave bands) |
| $L_{Vc_l}, u_{c_l}$ | Background-corrected SPL and corresponding uncertainty (aggregation over all microphones $m$ within partition $l$) |
| $L_{WA,\{v,\beta\}}, u_{L_{WA,\{v,\beta\}}}$ | Turbine apparent sound power level and corresponding uncertainty (combination over all relative positions $r$) |

### 3.3.2 Calculation of noise levels

Likewise the background noise, the total noise dataset is binned by wind speed for each SLM. However compared to the IEC standard two additional variables must be taken into account: one is of course the yaw offset angle applied to the turbine and the second is the relative position of the SLM with respect to the wind direction. Indeed, the noise pattern emitted by a turbine

is not homogeneous and depending on the wind direction, the SLM will not measure the same sound pressure level. For each 10-seconds chunk, the relative position of a SLM was calculated as the difference between the reference wind direction and the azimuth between the turbine and the SLM, and rounded to the closest 15° bin. Consequently it is no longer required to filter data when wind direction is changing by more than 15° as recommended in the IEC standard, the data are simply redispatched between the 24 SLMs of the circle.

Still, the supplementary partitioning added by this new binning variable complexifies the data processing. Due to the curse of dimensionality, with these 4 binning variables some of bins find themselves very sparsely populated. Accounting at this stage for the minimum number of 10 samples in each bin would discard a very large number of them (approximately 45 %). Alternatively, to retain as much data as possible, a much smaller value of 3 samples per bin was used. Then the formula of





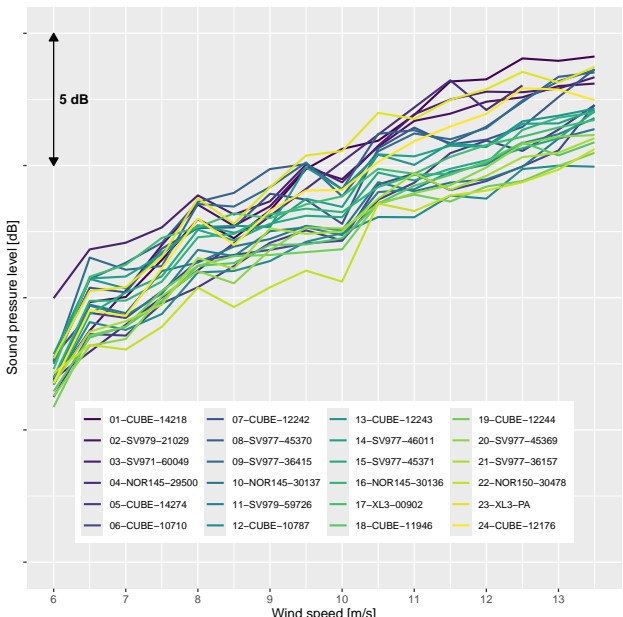

**Figure 11.** Evolution of median background noise as a function of wind speed bin for each SLM surrounding the wind turbine.

**Table 3.** Type B measurement uncertainties

| Component | Chosen value | Remark |
|---|---|---|
| Calibration, $u_{B1}$ | 0.2 dB | IEC 61400-11 proposed value |
| Instrument, $u_{B2}$ | 0.2 - 0.8 dB | Frequency dependent, based on a calibration certificate |
| Board, $u_{B3}$ | 0.3 dB | IEC 61400-11 proposed value |
| Wind screen, $u_{B4}$ | 0.2 - 0.9 dB | Frequency dependent, based on a calibration sheet |
| Distance and direction, $u_{B5}$ | 0.1 dB | IEC 61400-11 proposed value |
| Air absorption, $u_{B6}$ | 0 dB | IEC 61400-11 recommendation |
| Weather conditions, $u_{B7}$ | 0.5 dB | IEC 61400-11 proposed value |
| Wind speed, measured, $u_{B8}$ | 0.7 m/s | IEC 61400-11 proposed value |
| Wind speed, derived, $u_{B8}$ | 0.2 m/s | IEC 61400-11 proposed value |
| Wind speed, power curve, $u_{B9}$ | 0.2 m/s | IEC 61400-11 proposed value |

the IEC noise standard were applied to every bin in order to calculate the average wind speed and sound pressure level in
each 1/3-octave bands, evaluate the corresponding uncertainties and covariance, and interpolate the noise levels at bin centre
including uncertainties. The Tab. 3 summarizes the values that were taken for the type B uncertainties.





Next, the binned total noise noise is compared to the binned background noise for each microphone. Again, the process indicated by the standard was followed to obtain the background corrected sound pressure level at bin centre for each 1/3-octave band $i$ within each partition of wind speed $v$, SLM $m$, relative position $r$ and yaw offset $\beta$. The number of samples $N_k$ associated to each partition $k$ (where $k = \{v; \beta; r; m\}$) is taken as the lowest number of samples between the total noise and background noise datasets.

The end of the data reduction procedure differs from the one developed in the standard. Rather than calculating the sound power level for each 1/3-octave band, the background corrected overall A-weighted sound pressure level (OASPL) $L_{Vc_k}$ is calculated for each $k$ by energy summing of all 1/3-octave band background corrected sound pressure values $L_{Vc_{i,k}}$.

$$L_{Vc_k} = 10 \cdot \log \left( \sum_{i=1}^{28} 10^{\left( \frac{L_{Vc_{i,k}}}{10} \right)} \right) \tag{1}$$

For SLM with missing 1/3-octave bands, the measured OASPL was directly used. For those with incomplete 1/3-octave bands, the energy summation is done on the available bands, given that they were previously normalized to match the measured overall sound pressure level. The uncertainty $u_{c_k}$ associated to the obtained OASPL is deduced from the uncertainty of each 1/3-octave pressure value $u_{c_{i,k}}$ :

$$u_{c_k} = \frac{\sum_{i=1}^{28} u_{c_{i,k}} 10^{\left( \frac{L_{Vc_{i,k}}}{10} \right)}}{\sum_{i=1}^{28} 10^{\left( \frac{L_{Vc_{i,k}}}{10} \right)}} \tag{2}$$

Finally, the OASPL calculated for several microphones for the same relative position and yaw category are combined into a single value. This is done by computing a weighted average of $L_{Vc_k}$ values over all microphones within the same partition $l = \{v; \beta; r\}$:

$$L_{Vc_l} = 10 \cdot \log \left( \frac{\sum_{m=1}^{M_l} w_k 10^{\left( \frac{L_{Vc_k}}{10} \right)}}{\sum_{m=1}^{M_l} w_k} \right) \tag{3}$$

with the corresponding uncertainty

$$u_{c_l} = \sqrt{\frac{1}{\sum_{m=1}^{M_l} w_k}} \tag{4}$$

where $M_l$ is total number of microphones in the partition $l$, and the weights $w_k$ are chosen to minimize the variance of the derived estimator (Shahar, 2017):

$$w_k = \frac{1}{u_{c_k}^2} \tag{5}$$

The total number of samples associated to the partition $l$ is simply the sum of samples for all partitions $k$ :

$$N_l = \sum_{m=1}^{M_l} N_k \tag{6}$$





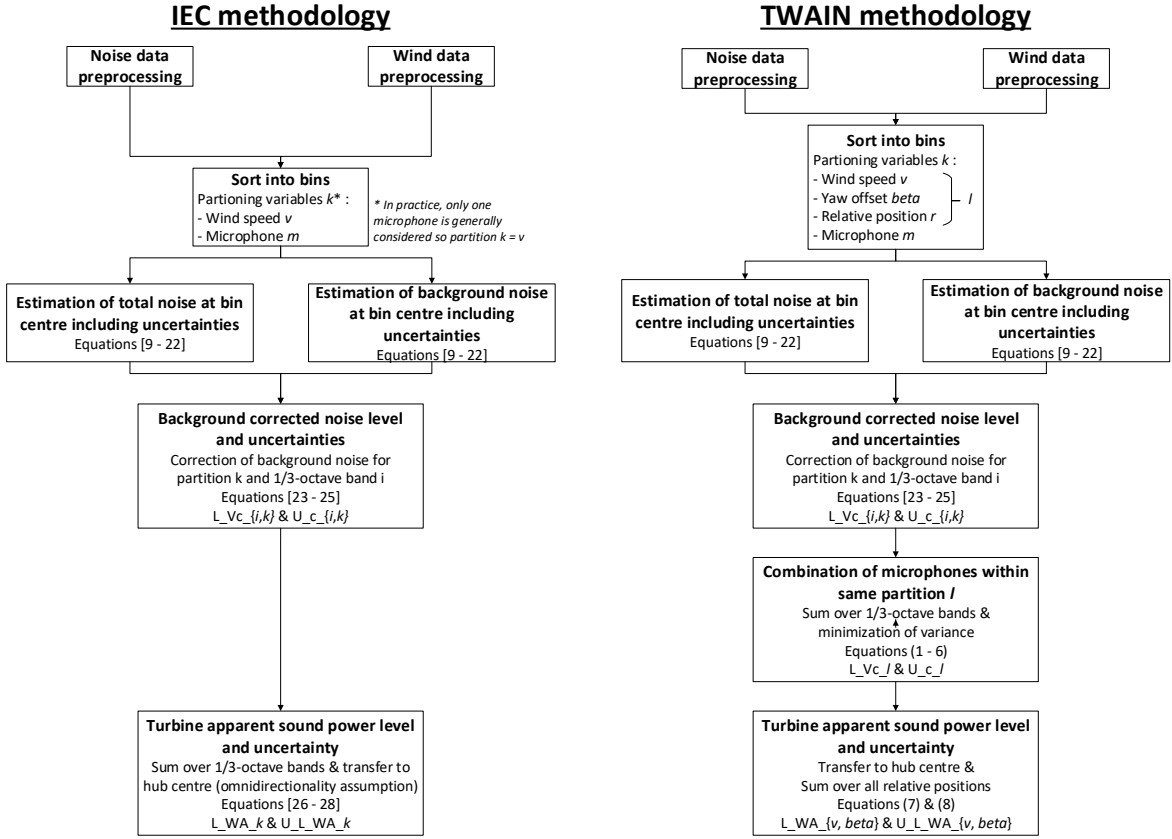

**Figure 12.** Comparison of data reduction procedure methodology developed in the IEC noise measurement standard (IEC et al., 2012) (left) and the new methodology proposed for the TWAIN experimental campaign (right). Numbers in brackets [.] refer to equations of the IEC standard, and the ones in parenthesis (.) correspond to equations in this present paper.

Only the partitions with $N_l \geq 10$ are kept for the end of the analysis. Thus, it fulfills the IEC constraint that at least 10 samples must be present in each bin, only that it is allowed that samples can come from multiple sensors to keep as much data as possible. With the applied process, microphones with the lowest uncertainty $u_{c_k}$ (generally because they have more samples in their partition) will have a stronger weight $w_k$ and will contribute more to the derived OASPL value.

The diagram shown in Fig. 12 recaps the data processing and underlines the differences with the current version of IEC 61400-11 standard.





### 3.3.3 Aggregation from noise levels to turbine apparent sound power level

Once the sound pressure values have been obtained for each relative position, they can be aggregated together to derive the
turbine apparent overall A-weighted sound power level (OASWL) for each wind speed and yaw position $L_{WA,\{v,\beta\}}$. Contrary
to the IEC standard that considers a single microphone located downstream with an omnidirectionality assumption for the noise
emission, this novel experimental setup allows for more finesse by accounting for the azimuthal dependence in the estimation
of acoustic power. The single contributions at each relative position around the turbine are energy summed, with the hypothesis
that the acoustic level is constant over a slice of the hemisphere with an area $4\pi R_1^2 \tau$, where $R_1 = 161.1$ m is the average distance
between all microphones and the rotor centre, and $\tau = 1/24$ is the angular slice ratio associated to each relative position $r$. This
yields the following equations for $L_{WA,\{v,\beta\}}$ and its respective uncertainty $u_{L_{WA,\{v,\beta\}}}$ (Finez et al., 2025):

$$L_{WA,\{v,\beta\}} = 10 \cdot \log \left( \sum_{r=1}^{24} 10^{\left( \frac{L_{Vc_l} - 6}{10} \right)} \frac{4\pi R_1^2}{S_0} \tau \right) \tag{7}$$

$$u_{L_{WA,\{v,\beta\}}} = \frac{\sum_{r=1}^{24} u_{c,l} 10^{\left( \frac{L_{Vc_l} - 6}{10} \right)} \frac{4\pi R_1^2}{S_0} \tau}{\sum_{r=1}^{24} 10^{\left( \frac{L_{Vc_l} - 6}{10} \right)} \frac{4\pi R_1^2}{S_0} \tau} \tag{8}$$

where $S_0 = 1$ m$^2$ is a reference area. A linear interpolation from adjacent observer positions $r$ is operated in case of missing
$L_{Vc_l}$ values due to a lack of measurement data. If more than 4 points out of the 24 positions were missing for a given wind
speed and yaw offset case, the corresponding OASWL value was not calculated.

With Eq. 7, the effect of yaw misalignment on turbine apparent sound power can be properly estimated, without being biased
by the rotation of the noise directivity pattern, as will be discussed in the next section.

## 4 Results and discussion

Following the data processing methodology described in Sect. 3, noise directivity patterns are calculated on the measurement
data and presented in this section. In order to harmonize the results coming from multiple wind directions, the sign conventions
defined on Fig. 13 are adopted. First, turbine noise emissions are analyzed when it is aligned with the wind in Sect. 4.1, then
the effect yaw misalignment is compared to the no yaw situation in Sect. 4.2.

### 4.1 Directivity of turbine noise emissions with no misalignment

OASPL of the estimated turbine noise are presented in Fig. 14 for each observer position relative to the nacelle axis without
yaw misalignment and across four wind speed bins. Since the turbine reaches rated power at 12 m/s, the selected wind speeds
correspond to 52% to 100% of rated power. At all observed wind speeds, OASPL variations in the ground-based directivity
pattern reach up to 4 dB(A). In each plot, a region of low noise levels is observed near the rotor plane, sometimes referred to as
"noise dips" in the literature (Oerlemans and Schepers, 2009). Within these dips, sound levels are consistently 3.5 to 4.0 dB(A)





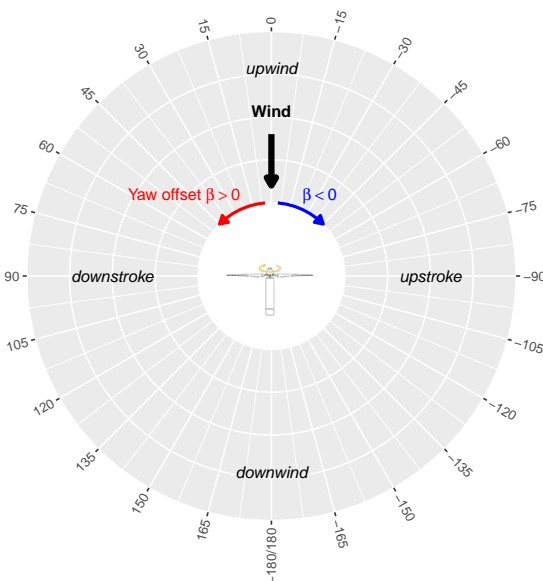

**Figure 13.** Sign convention followed for the display of the directivity patterns as a function of the observer angle $\theta$. Wind direction is always coming from the angle $\theta = 0°$, while $\theta = -180°/180°$ is the downwind position. Positive yaw angles $\beta$ are defined as a counter-clockwise rotation of the nacelle when seen from above. Positive and negative relative positions corresponds to the downstroke and upstroke side of the rotor plane, respectively.

lower than the maximum values in the corresponding directivity pattern. At lower speeds (7.5 and 9.5 m/s), the minimum noise levels occur at observer angles of $\theta = \pm90°$. Interestingly near rated power (11.5 m/s and 13.5 m/s), the noise dips shift upstream, appearing at $\theta = \pm75°$.

The direction of the maximum noise level is not aligned with the downstream flow axis, but rather occurs at off-axis positions, either upstream or downstream of the turbine. Specifically, the peak noise levels are observed at $\theta = -150°$ at 7.5 m/s, $\theta = 135°$ at 9.5 m/s, $\theta = 45°$ at 11.5 m/s and $\theta = 120°$ at 13.5 m/s. These levels exceed the axial downstream measurement by 1.4, 0.7, 0.8 and 1.6 dB(A) respectively. This observation is not captured by most existing analytical and semi-empirical models (Bertagnolio et al., 2023) which typically predict symmetric directivity patterns with the maximum aligned along the main flow axis. In contrast, the directivity patterns shown in Fig. 14 exhibit slight asymmetries, not only with respect to the nacelle axis, but also relative to the rotor plane. Since the observed differences in noise levels remain within the uncertainty bounds, a statistical paired t-test was conducted to support this visual observation. The test is based on the level difference between two symmetrical positions $\Delta L_{V,\theta} = L_{V,\theta} - L_{V,\theta'}$ for the same wind speed bin, where $\theta'$ is chosen as:

–  $\theta' = -\theta$ to test the down-upstroke symmetry and

–  $\theta' = 90° - \theta$ to test the down-upwind symmetry.



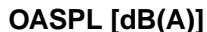

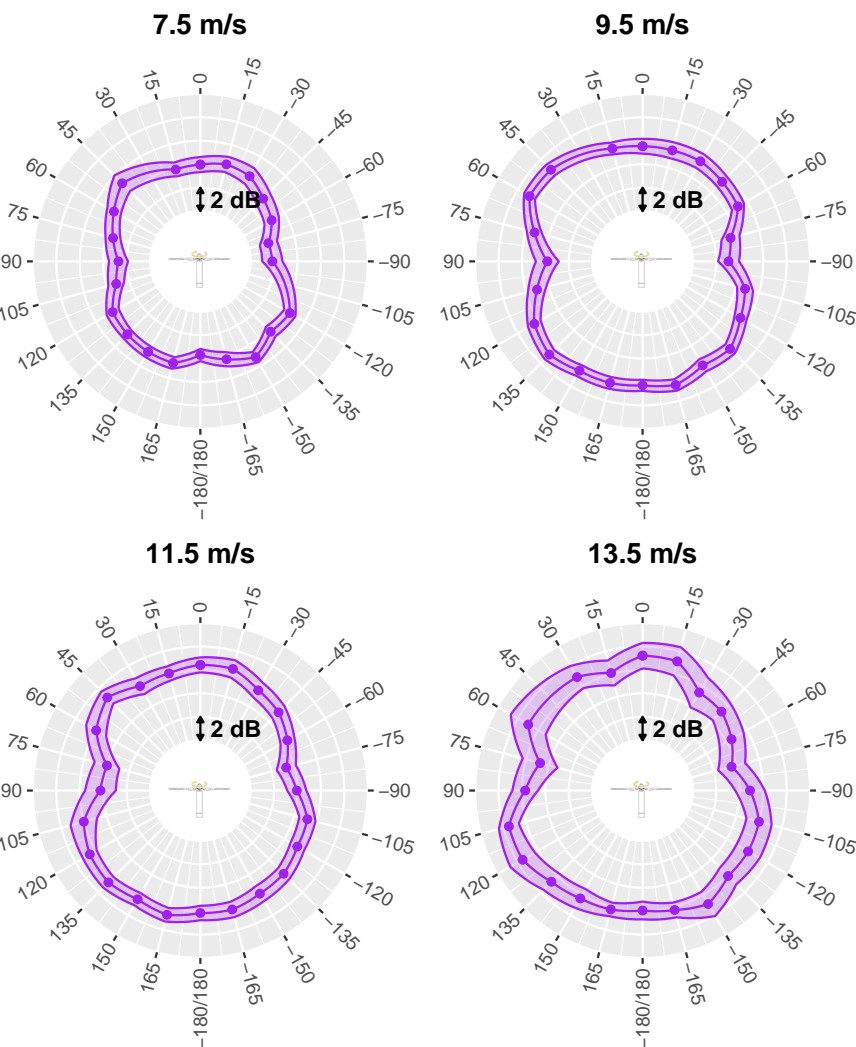

**Figure 14.** Directivity of overall A-weighted sound pressure level (OASPL) for 4 different wind speeds. In each graph, the wind is coming from the top (position 0°), positive and negatives angles corresponds to downstroke and upstroke side of the rotor, respectively. The shaded ribbon represents one standard uncertainty associated to each OASPL value $u_{c_l}$, while the point markers indicate the mean value $L_{Vc_l}$. For observer positions containing less than 10 valid samples, the corresponding data point is omitted.

The set of available observation angle pairs $(\theta; \theta')$ is noted $\Theta$ and consists of 207 observations in both of the above mentioned cases. The level difference $\Delta L_{V,\theta}$ is associated with a combined variance $u_\theta^2$ such as

$$u_\theta^2 = u_{c,\theta}^2 + u_{c,\theta'}^2 - 2\rho\, u_{c,\theta}\, u_{c,\theta'} \tag{9}$$





where $\rho$ is the correlation factor between the $\theta$ and $\theta'$ series, which may be non-zero due to the previously identified noise dips. Following the approach in Eqs. 3 - 5, the weights are defined as the inverse of the combined variance $w_\theta = 1/u_\theta^2$ and the weighted average of the level difference is computed as:

$$\overline{\Delta L_V} = \frac{\sum_{\theta \in \Theta} w_\theta \Delta L_{V,\theta}}{\sum_{\theta \in \Theta} w_\theta} \tag{10}$$

with the standard error

$$\sigma_{\Delta L} = \sqrt{\frac{1}{\sum_{\theta \in \Theta} w_\theta}}. \tag{11}$$

This enables to use the test statistics $t = \overline{\Delta L_V}/\sigma_{\Delta L}$ which is applied across the full set of available wind speed bins at once, using a significance level of 5% and assuming a normal distribution. On average, the OASPL value on the downstroke side is found to be 0.6 dB higher than on the upstroke side. The 95 % confidence interval is [0.5 ; 0.8] dB, associated to a p-value $p < 10^{-6}$, indicating a highly significant left/right asymmetry in the directivity pattern. This result was found to be virtually independant of the correlation factor $\rho$ within the range 0 to 0.5.

The same approach was applied to assess the symmetry between downwind and upwind directions. On average, the sound level measured on the downwind side was 0.4 dB higher than that on the upwind side, at observer positions symmetric with respect to the rotor plane. The 95% confidence interval for this difference is [0.3; 0.6] dB(A), associated to a p-value $p < 10^{-8}$, indicating again a highly significant asymmetry.

From this analysis, it can be concluded that the directivity footprint of the turbine exhibits, on average, higher noise levels on the downstroke side compared to the upstroke side by 0.6 dB(A), and higher levels on the downwind side compared to the upwind side by 0.4 dB(A). These slight asymmetries in the turbine's noise radiation may result from the intrinsic directivity of elementary acoustic sources. Analytical models based on flat plate assumptions (as for instance Tian and Cotté (2016)) as well as (semi-)empirical models derived from symmetric airfoil measurements (Bertagnolio et al., 2023) do not reproduce such asymmetries. However, camber has been shown to cause deviations of up to 1 dB(A) between the pressure and the suction side noise radiation of an non symmetrical airfoil (Roger and Moreau, 2010). This behavior has also been captured in high-fidelity numerical simulations of cambered and loaded airfoils equipped with serrations (Romani et al., 2021).

Spectral directivity patterns at near-rated power (11.5 m/s) are presented in Fig. 15 over a selected frequency range. At very low frequencies (e.g., 50 Hz), the pattern exhibits two distinct lobes oriented upstream and downstream, with higher levels on the downstream side. Pronounced noise dips are also observed, with sound levels up to 6 dB lower than at the axial downwind position, which corresponds to the direction of maximum noise emission at this frequency. As frequency increases, the directivity pattern becomes more complex, featuring several off-axis lobes. For instance, in the mid-frequency range around 1000 Hz—where human hearing is most sensitive—a narrow lobe centered at $\theta = 45°$ emerges, with noise levels approximately 4 dB higher than adjacent directions. A secondary, less pronounced lobe appears on the descending blade side around $\theta = 120°$. These features are also reflected in the OASPL directivity patterns (lower left part of Fig. 14) at $\theta = 45°$ and $\theta = 135°$. At higher frequencies (e.g., 5000 Hz), the pattern displays multiple lobes with significant local variations and increased uncertainty ($\pm 2.0$ dB), compared to lower frequencies.



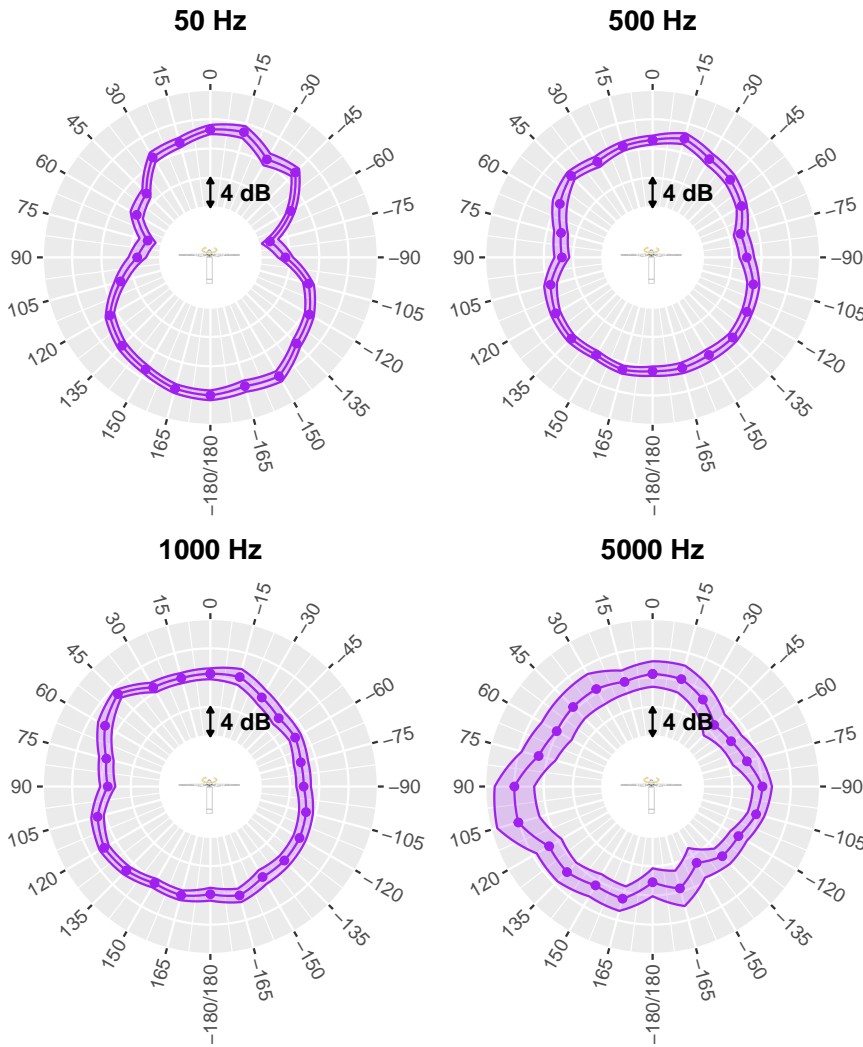

**Figure 15.** Directivity of overall A-weighted sound pressure level (OASPL) at 11.5 m/s for 4 different 1/3-octave bands, without yaw misalignment. The shaded ribbon represents one standard uncertainty.

A complementary perspective is provided in Fig. 16 which compares frequency spectra at selected observer positions. In Fig. 16 (a) the maxima of the two OASPL lobes at 11.5 m/s – located at $\theta = 45°$ and $\theta = 135°$ are compared to the reference axial downstream position ($\theta = 180°$). At these two positions, the spectral content shows a pronounced mid-frequency component between 800 Hz and 2000 Hz with increases of 5 dB and 3 dB respectively, relative to the reference. The very high frequency range (above 5000 Hz), further increases of +7 dB and +5 dB are observed. Conversely a reduction in the low-frequency range (50 Hz - 500 Hz) is noted, particularly at $\theta = 45°$, where levels are 7 dB(A) lower than the reference.




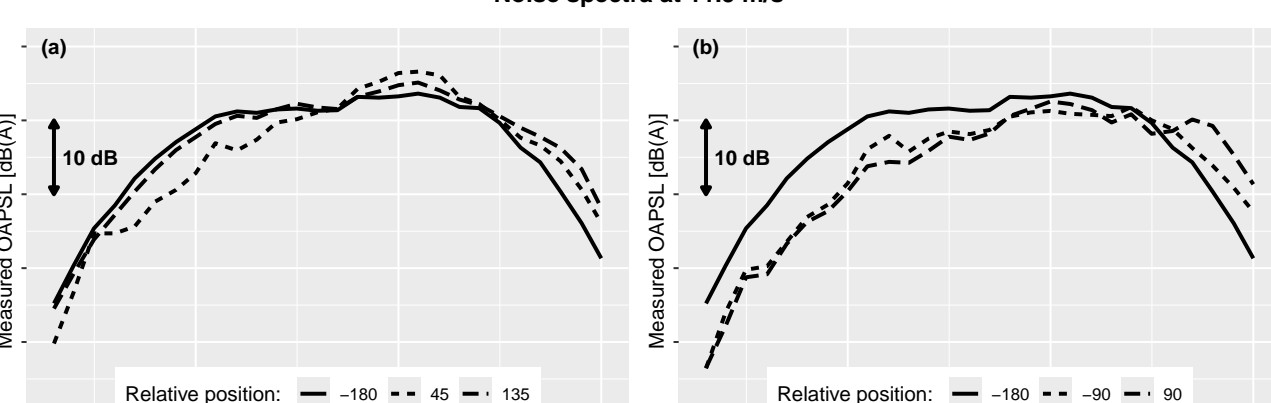

**Figure 16.** Noise spectra measured at 11.5 m/s for different positions of the circle. On the left (a), spectra at the two lobes of the directivity pattern (45° for the upstream lobe, 135° for the downstream) are compared to the spectrum at central downstream location (180°). On the right (b), spectra in the rotor plane (90° for upstroke, -90° for downstroke) are compared to the downstream location (180°).

Figure 16 (b) presents two additional measurement points at the same wind speed, comparing positions in the rotor plane to the axial downwind reference. In the noise dip regions, the low-frequency content is significantly reduced – by up to 10 dB at 100 Hz. In contrast, an increase in the very high-frequency range is observed, especially on the descending blade side 405 ($\theta = 90°$). In summary, at this specific wind speed, the spectral analysis reveals a strong broadband component in the mid-frequency range at off-axis lobes maxima, as well as a spectral shift toward higher frequencies in the noise dip regions, relative to the axial downwind position.

## 4.2 Effect of yaw misalignment

Directivity patterns under intentional yaw misalignment are illustrated in Fig 17 at 9.5 m/s and 11.5 m/s. The first row shows 410 positive yaw offsets (i.e., the ascending blade is oriented toward the incoming flow), while the second row presents negative yaw offsets (i.e., the ascending blade is oriented away from the flow). A first observation is that yaw misalignment does not lead to dramatic changes in overall noise levels. The most prominent effect of steering the turbine away from the main flow direction is a rotation of the entire directivity pattern, which appears to remain aligned with the turbine orientation in all four cases – particularly around the recognizable noise dip regions. It is worth noting that the yaw angle sampling step $\Delta\beta = 10°$ is 415 not a multiple of the observer angle sampling step $\Delta\theta = 15°$, which explains why the noise dips may appear at similar positions for different yaw angles. Some regions appear more strongly influenced by the flow direction than by the turbine orientation and are amplified when a yaw offset is applied. This is particularly evident in the upwind-upstroke sector $[0°; -45°]$ for $\beta > 0$

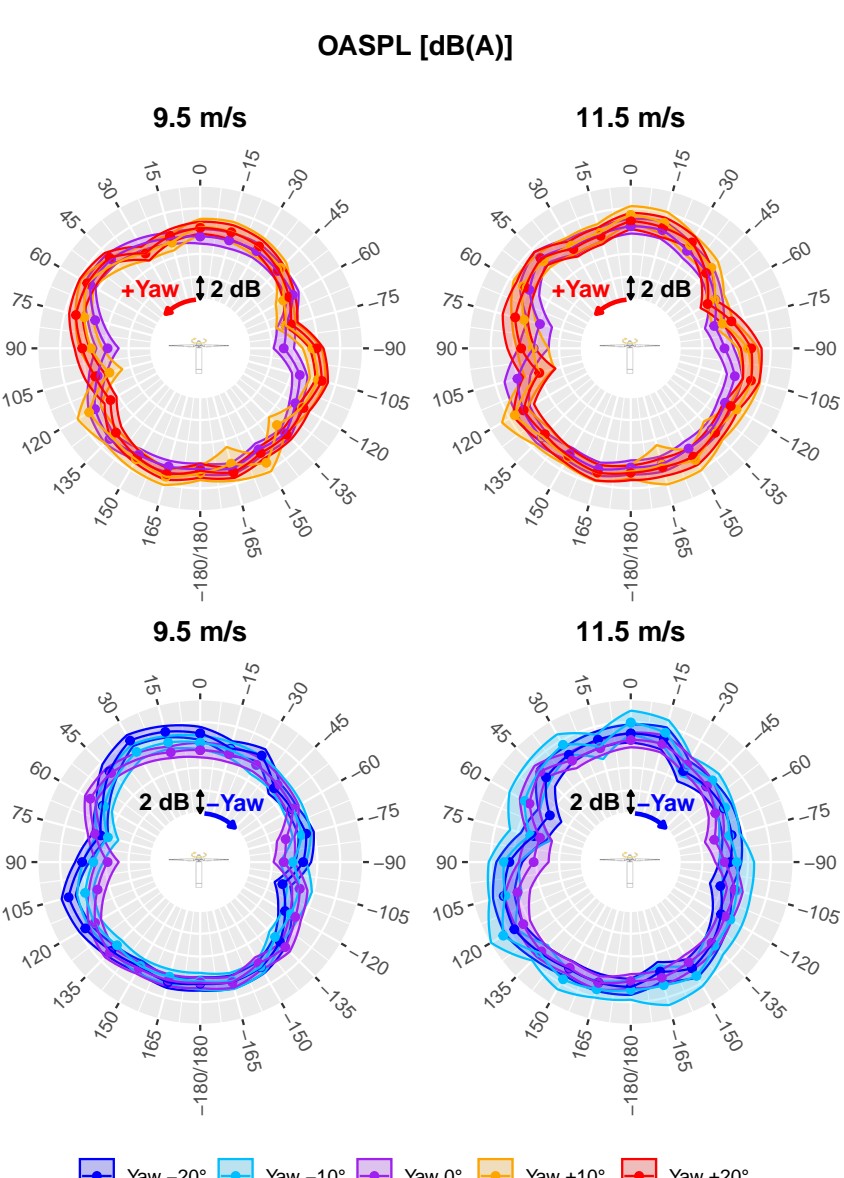

**Figure 17.** Directivity of overall A-weighted sound pressure level (OASPL) for positive (top) and negative (bottom) yaws, and 2 different wind speeds. In each graph, the wind is coming from the top (position 0°), positive and negatives angles corresponds to downstroke and upstroke side of the rotor, respectively. The shaded ribbon represents one standard uncertainty associated to each OASPL value $u_{c_l}$, while the point markers indicate the mean value $L_{Vc_l}$. For observer position containing less than 10 valid samples, the corresponding data point is omitted.

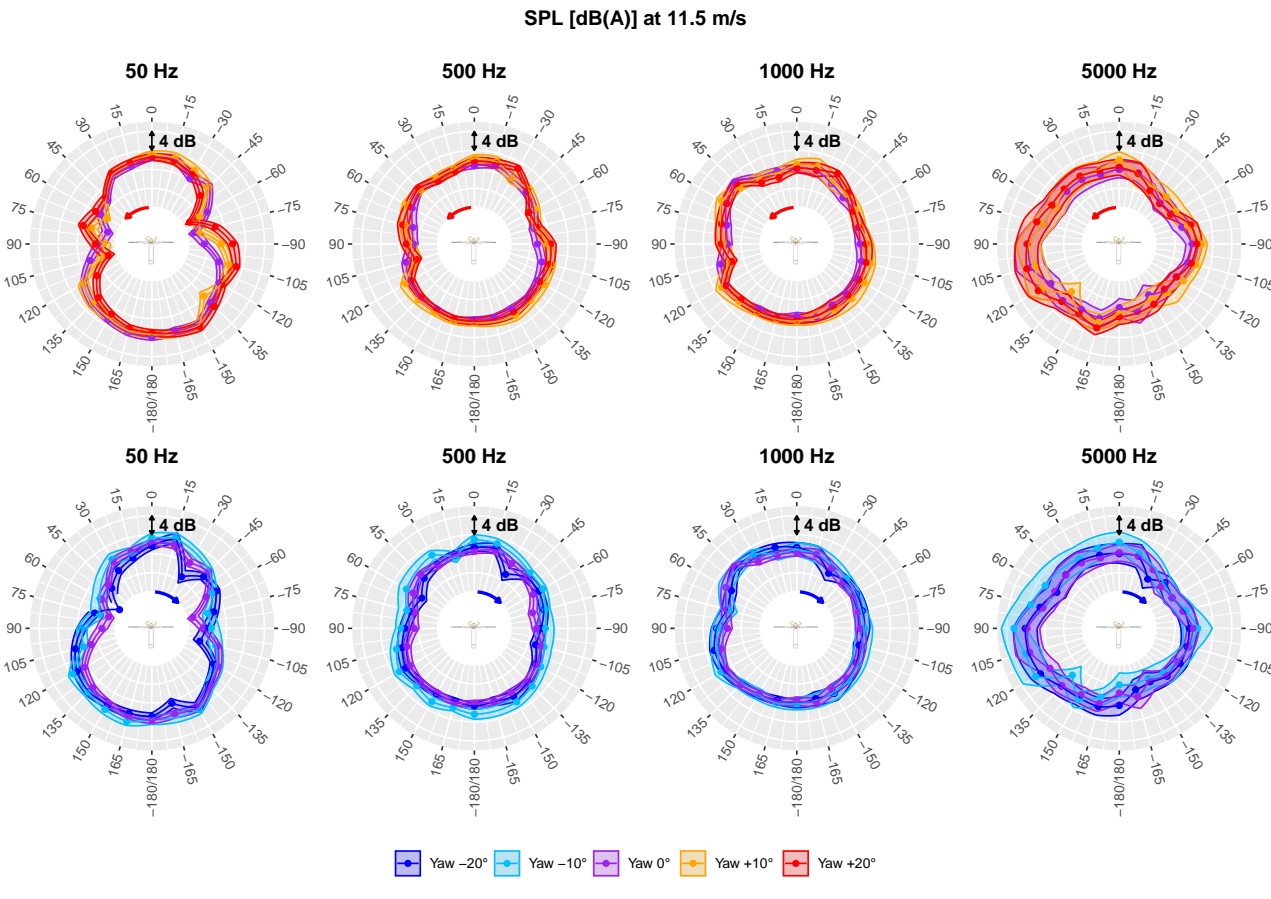

**Figure 18.** Directivity of A-weighted sound pressure level (SPL) at 11.5 m/s for positive (top) and negative (bottom) yaws, and 4 different 1/3-octave bands. The shaded ribbon represents one standard uncertainty.

and in the upwind-downstroke sector $[0°; +45°]$ for $\beta < 0$. In general, larger absolute yaw offsets $\beta$ tend to introduce greater complexity into the directivity pattern.

Spectral variations in the directivity patterns at $v = 11.5$ m/s are illustrated in Fig. 18. At the lowest frequencies (50 Hz), the rotation of the directivity pattern induced by the turbine misalignement is particularly visible. The most significant variations with respect to the yaw angle $\beta$ are predominantly localized near the rotor plane ($|\theta| \simeq 90°$). The acoustic impact of yaw offset is more evident at the lower (50 Hz) and upper (5000 Hz) ends of the frequency spectrum, compared to the mid-range frequencies (500 Hz and 1000 Hz). This behavior may be attributed to the relative variations of different noise generation

mechanisms: turbulent inflow noise, which usually dominates at low frequencies, and turbulent boundary layer – trailing edge noise, which is prominent at high frequencies. These distinct sources may be differently affected by the yaw misalignment.





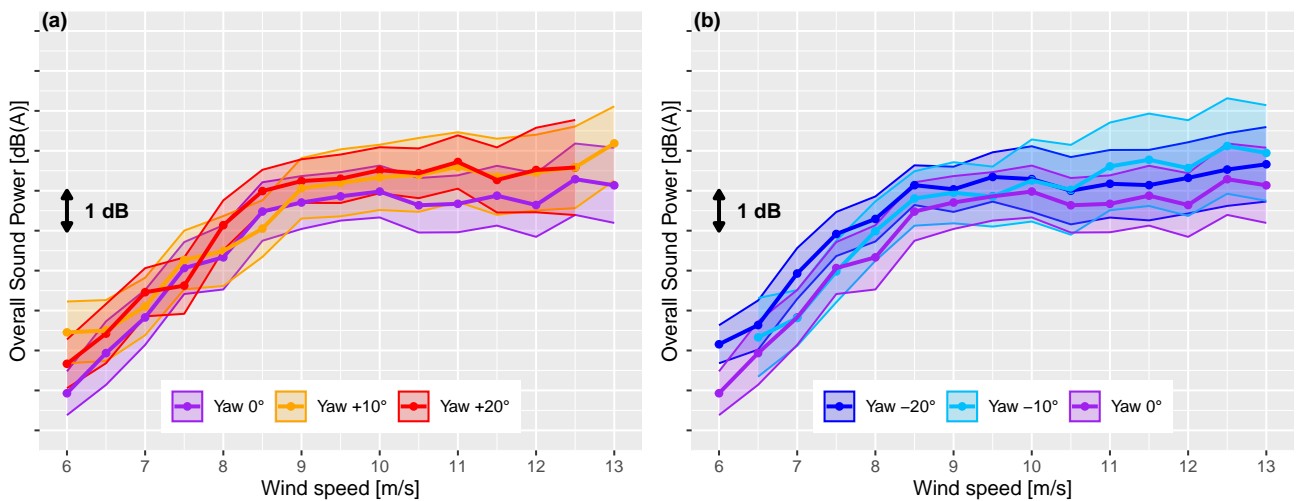

**Figure 19.** Turbine sound power level (OASWL) as a function of wind speed for positive (a) and negative (b) yaws. The shaded ribbon represents one standard uncertainty associated to each OASWL value $u_{L_{WA,\{v,\beta\}}}$, while the point markers indicate the mean value $L_{WA,\{v,\beta\}}$. For wind speed bins and yaw offset cases with too few observer positions, the point is omitted.

Estimates of turbine's OASWL derived using Eqs. 7 and 8 are presented in Fig. 19 for various yaw offset angles, both positive (left panel) and negative (right panel). All empirical sound power curves exhibit a consistent structure, characterized by a linear increase between 6 m/s and 9 m/s, followed by a plateau at higher wind speeds, as expected for this turbine. For most wind speed bins, the aligned configuration $\beta = 0°$ is the less noisy configuration. In contrast, configurations with non-zero yaw offsets tend to exhibit more elevated sound power levels. To evaluate the statistical significance of these observations, four inverse-variance weighted paired t-tests were conducted using Eqs. 10 and 11. Each test compares the sound power curve for a misaligned configuration $L_{WA,\{v,\beta\neq0°\}}$ against the reference aligned configuration $L_{WA,\{v,\beta=0°\}}$. The result, summarized in Tab. 4, indicate that for yaw angles of $\beta = -20°, +10°, 20°$, the differences are statistically significant ($p < 0.05$). These configurations exhibit a consistent increase in OASWL of approximately 0.5 to 0.7 dB(A). Accordingly, it can be concluded that these misaligned settings are slightly noisier than the baseline configuration, with an average increase in sound power of approximately 0.6 dB(A).

The slice summation described in Eq. 7 can be independently computed for each 1/3-octave band, yielding the turbine's sound power level (SWL) spectra. These spectra are presented in Fig. 20 for various yaw offset configurations. Consistent with the OASPL analysis, all yawed configurations exhibit higher spectral levels compared to the baseline (aligned) case. The increase is particularly notable in the low to mid-frequency range (200 – 1250 Hz) as well as in the high-frequency range ($f > 5$ kHz). As previously discussed in the analysis of Fig.18, this pattern suggests that the dominant noise generation mechanisms are differentially influenced by turbine yaw misalignment.



**Table 4.** T-test results for the offset in sound power level curves compared to the aligned case.

| Yaw offset $\beta$ (°) | $\overline{\Delta L_{WA}}$ (dB(A)) | 95% Confidence Interval (dB(A)) | p-value |
|---|---|---|---|
| -20 | +0.7 | [0.25;1.21] | 0.003 |
| -10 | +0.5 | [-0.08; 1.09] | 0.09 |
| 10 | +0.6 | [0.07;1.12] | 0.03 |
| 20 | +0.6 | [0.14;1.13] | 0.01 |

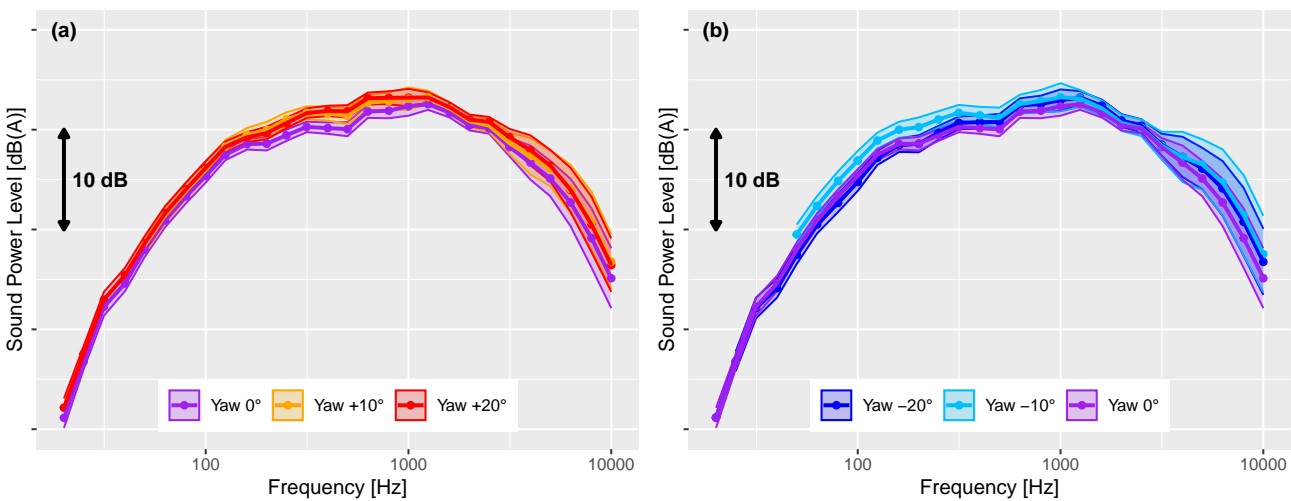

**Figure 20.** Turbine sound power level spectra at $v = 11$ m/s for positive (a) and negative (b) yaws. The shaded ribbon represents one standard uncertainty.

The results presented here may be compared to the extensive noise measurement campaign conducted by Hamilton et al.

(2021) near a 77 m, diameter 1.5 MW industrial wind turbine submitted to a yaw offsets ranging from -18° to 25°. Their study focused on the downstream region of the turbine with a viewing angle of approximately $100°$. A comparison of noise levels at *fixed* ground positions with and without yaw generally revealed a slight decrease of 1 to 2 dB at moderate yaw angles. In contrast, the present study observed a slight increase of +0.6 dB in noise power levels under yaw conditions. This discrepancy may be attributed to differences in turbine model, size and control system. Additionally, a partial view on the directivity pattern

and its rotation may result in local sound pressure level decreases which may be attributed to an apparent sound power level decrease.





# 5  Conclusions

A novel acoustic measurement campaign was conducted around an industrial 2.2 MW wind turbine, involving 24 ground-based sound level meters and multiple lidar devices to monitor incoming wind speed and direction. The experiment encompassed a wide range of wind conditions, enabling the construction of detailed directivity patterns across the turbine's full operational range. Modifications to the IEC standard methodology were proposed to leverage the advantages of this multi-point measurement setup. Furthermore, the turbine was operated both under standard alignment control and with intentional yaw offsets (ranging from $-20°$ to $+20°$), in order to investigate sound power variations associated with wake steering strategies.

Measurements conducted under non-steered conditions reveal that the turbine generally exhibits a two-lobe directivity pattern aligned with the nacelle axis. However, in most tested configurations, the axial downwind direction does not correspond to the direction of maximum emissivity, with differences ranging from 0.7 to 1.6 dB(A), contrary to predictions from most semi-empirical and analytical wind turbine noise models. Furthermore, the measured directivity patterns display asymmetry, with a statistically significant offset of +0.6 dB on the downstroke side compared to its symmetrical upstroke counterpart, and +0.4 dB on the downwind side relative to the upwind side. Spectral comparisons in the off-axis directions of maximum OASPL ($\theta = +45°$ and $\theta = +135°$) reveal a pronounced broadband contribution around 1000 Hz, reduced sound pressure levels at low frequencies, and elevated levels at high frequencies ($f > 5$ Hz).

Acoustic curtailment plan are currently still derived assuming turbines are omnidirectional noise sources. By accounting for a more detailed directivity pattern in the design of those plans, wind farm operators could be able to develop much refined strategies that could in turn lead to significant energy gains over the full lifetime of a project.

When a yaw offset is applied, the directivity pattern rotates in alignment with the turbine's axis, exhibiting increased complexity with additional lobes and dips compared to standard operating conditions. An increase in the turbine's sound power level (SWL), estimated from the multi-point ground-based acoustic footprint, is observed -— approximately +0.6 dB(A), independently of the actual yaw angle value. This increase is particularly pronounced in the low-frequency range (below 1000 Hz) and the high-frequency range (above 5000 Hz), suggesting that yaw misalignment differentially affects the primary noise generation mechanisms.

This first of his kind experiment underlines the complexity of multi-objective WFC and the need for more detailed analytical noise models. Indeed, while wake steering has shown consistent results for mitigating wake effects and improving overall wind farm production, the increased noise emissions could lead to more constraining acoustic curtailment plan that will have the opposite effect on farm performance. Reliable models must be developed so that operators can weight the costs and benefits of their choices, and make better informed decision. Future work in the scope of the TWAIN project will focus on the validation of more advanced turbine noise emission models using the data generated through this experiment, which will later be used in a case study combining wake steering and noise optimization.

*Data availability.* Since the experiment was realized on a commercial wind turbine, the data used in this research cannot be made available.



*Author contributions.* The TWAIN acoustic campaign was designed and realized by TD and AF. Wind related data were post-processed by
TD and acoustic related data by AF. The methodology detailed in Sect. 3 was derived conjointly by TD and AF. TD wrote sections 1, 2 and 3 and AF wrote sections 4 and 5. Both authors reviewed and edited the manuscript.

*Competing interests.* The authors declare that they have no conflict of interest.

*Acknowledgements.* The TWAIN acoustics campaign is realized under the TWAIN project, funded by the European Union's Horizon Europe research and innovation program with grant agreement no. 101122194.
We would like to thank Florent Bruneau and all the Echopsy team for their great support in setting up and conducting the field test. Additionally, we would like to acknowledge four other ENGIE Green colleagues: Warren Herbaut for solving all operational constraints related to the preparation of the campaign, Assia Achhibat for the help in the data preprocessing, Paul Mazoyer for the tip allowing us to properly combine data from multiple microphones at a given relative position and Colin Le Bourdat for the general support of this work.



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
