# Peer review of "Experimental investigation of the effect of wake steering on the noise emission of a commercial wind turbine"

_Wind Energy Science, 2025_

## Referee Comment (RC1)

**Peer review for WES-2025-224**

**Title**: "Experimental investigation of the effect of wake steering on the noise emission of a commercial wind turbine"

**Authors**:

- Thomas Duc
- Arthur Finez

**Overview**

This manuscript investigates the effect of yaw misalignment on wind turbine noise emissions through a full-scale acoustic measurement campaign. The experimental design addresses a relevant knowledge gap: quantifying acoustic implications of wake steering strategies increasingly deployed for power optimization. The study reports statistically significant directivity asymmetries that challenge omnidirectional noise source assumptions in current acoustic models, and observes a modest but consistent increase of approximately 0.6 dB(A) in overall sound power level under yaw misalignment. While the experimental approach is innovative and the dataset appears comprehensive, the manuscript suffers from insufficient justification of key methodological choices, inadequate discussion of contradictions with prior literature, and presentation issues that limit accessibility of the results.

The work represents a valuable contribution to the WFFC literature and provides high-quality data for model validation. The experimental design and dataset quality are strong, but methodological transparency, contradiction with prior literature, and presentation quality should be improved before publication. The authors should provide justification for processing parameter choices and discussion reconciling the current results in the context of past work in wind turbine aeroacoustics and directivity.

**Major Concerns**

1. The observed increase in sound power under yaw misalignment appears to contradict past literature cited by the authors, which reported noise reductions or no significant effect from yaw misalignment. The authors cite these studies but offer no substantive discussion of why the current results diverge. Are the differences from turbine model, blade aerodynamics, study or measurement methodology?

2. Time shifts of up to ±15 seconds are substantial relative to atmospheric turbulence timescales and microphone sampling frequency. The manuscript does not explain the root cause of such large clock drift over a four-day campaign. The cross-correlation procedure reduces uncertainty to approximately 1 second, but consequences for 10-second aggregated acoustic measurements remain unquantified. Given that acoustic propagation times span ~1 second across the measurement domain, this uncertainty may not be negligible for directivity pattern analysis. Can the authors speculate on the impacts of the time delays as a source of measurement uncertainty? How does this influence the final results?

3. Multiple filtering and detection parameters appear arbitrary without justification. The ±50 second rolling median window and 10 dB exceedance threshold for outlier detection (Section 3.1.2) lack supporting rationale. What is the false positive rate? How many genuine turbine noise events are erroneously filtered? Have the authors checked the manually recorded noise events against the automated event detection? The reduction of minimum samples per bin from 10 (IEC standard) to 3 (Section 3.3.2) to avoid discarding 45% of data is concerning.

4. The frequency-dependent nature of directivity changes (Figure 18, low-frequency effects dominate) and non-monotonic changes with wind speed (Figure 19, moderate yaw offsets produce larger changes than extreme offsets at high wind speeds) suggest complex underlying aeroacoustic physics that are not explained in the manuscript. What mechanisms separate low and high-frequency noise emission responses to yaw misalignment? Why do moderate yaw angles produce larger acoustic changes at high wind speeds? These patterns may reveal

fundamental insights into trailing-edge noise, turbulent inflow interactions, or blade loading asymmetries that should be discussed at least.

**Detailed Comments**

- Section 2.1, line 95: Was any testing conducted to quantify the acoustic impact of uncut versus cut wheat? Background noise measurements (Figure 11) show up to 4 dB variation across SLMs at fixed wind speed, attributed to "local environment" differences. Quantifying the wheat crop contribution would strengthen confidence that site-specific background variations are adequately characterized rather than introducing systematic bias in turbine noise extraction.

- Figures 2, 3, 6, 7: SLM identifiers should be consistent across all figures and text. Adopt a single convention (preferably the full identifier for traceability) and apply uniformly. Cross-check all figures to ensure labels match the site map.Axis labels, annotations, and colorbar text are difficult to read in several figures.

- Section 3.2, lines 220-237: The hybrid approach introduces potential discontinuities. Report the fraction of data falling into each category and assess whether transitions between wind speed sources introduce artifacts. The 1-minute rolling average applied to nacelle lidar data may smooth out turbulence-driven acoustic fluctuations; discuss whether this temporal filtering impacts correlation between inflow conditions and noise metrics.

- Figure 16: Consider combining separate panels into a single multi-panel figure with consistent axis ranges for direct comparison. Absolute axis limits would be extremely helpful in interpreting results. If permissible, please include.

- Figure 18: Changes in directivity induced by yawed operation may be more evident if results were presented in the rotor reference frame rather than the wind reference frame. Have the authors explored this alternative representation? The strong frequency dependence (low frequencies affected, high frequencies unaffected) suggests distinct noise generation mechanisms. What aeroacoustic processes explain the frequency threshold separating responsive and non-responsive regimes? Discuss whether low-frequency changes correlate with blade loading asymmetries, inflow turbulence modulation, or wake-induced effects.

- Figure 19: The current presentation obscures important trends. Plotting the change in noise emissions relative to the yaw 0-degree baseline as a function of wind speed would clarify how acoustic impacts vary across the operational envelope. The apparent result that moderate yaw offsets (±10 degrees) produce larger noise increases than extreme offsets (±20 degrees) at high wind speeds is counterintuitive and potentially significant. Does this non-monotonicity suggest optimal yaw angle strategies for minimizing acoustic impact while maintaining wake steering benefits? Provide mechanistic interpretation grounded in blade aerodynamics or turbulent inflow physics.

**Minor Edits and Technical Corrections**

- "Likewise" should be "Like" for grammatical correctness in this context.
- Ensure consistent terminology for key concepts (e.g., "yaw misalignment" vs "yaw offset" vs "yaw control") throughout the manuscript.
- Section 1, line 40: Remove extra comma in citation "(Göçmen, 2016), ."
- Section 2.3, line 125: "Monday 25th" should specify "25 March 2024" for clarity, as the month is mentioned only in the preceding paragraph.